# Antitumor Activity of Ruthenium(II) Terpyridine Complexes towards Colon Cancer Cells In Vitro and In Vivo

**DOI:** 10.3390/molecules25204699

**Published:** 2020-10-14

**Authors:** Maja Savic, Aleksandar Arsenijevic, Jelena Milovanovic, Bojana Stojanovic, Vesna Stankovic, Ana Rilak Simovic, Dejan Lazic, Nebojsa Arsenijevic, Marija Milovanovic

**Affiliations:** 1Department of Pharmacy, Faculty of Medical Sciences, University of Kragujevac, 34000 Kragujevac, Serbia; maja.jovanovic@medf.kg.ac.rs; 2Center for Molecular Medicine and Stem Cell Research, Faculty of Medical Sciences, University of Kragujevac, 34000 Kragujevac, Serbia; salvatoredjulijano@gmail.com (A.A.); jelenamilovanovic205@gmail.com (J.M.); bojana.stojanovic04@gmail.com (B.S.); arne@medf.kg.ac.rs (N.A.); 3Faculty of Medical Sciences, Institute of Histology, University of Kragujevac, 34000 Kragujevac, Serbia; 4Faculty of Medical Sciences, Institute of Pathophysiology, University of Kragujevac, 34000 Kragujevac, Serbia; 5Department of Pathology, Faculty of Medical Sciences, University of Kragujevac, 34000 Kragujevac, Serbia; wesna.stankovic@gmail.com; 6Department of Natural Sciences, Institute for Information Technologies Kragujevac, University of Kragujevac, Jovana Cvijića bb, 34000 Kragujevac, Serbia; anarilak@kg.ac.rs; 7Department of Surgery, Faculty of Medical Sciences, University of Kragujevac, Svetozara Markovića 69, 34000 Kragujevac, Serbia; dlazic.kg@gmail.com

**Keywords:** colon carcinoma, cytotoxicity, ruthenium(II) complexes, oxaliplatin

## Abstract

Ruthenium complexes have attracted considerable interest as potential antitumor agents. Therefore, antitumor activity and systemic toxicity of ruthenium(II) terpyridine complexes were evaluated in heterotopic mouse colon carcinoma. In the present study, cytotoxic effects of recently synthesized ruthenium(II) terpyridine complexes [Ru(Cl-tpy)(en)Cl][Cl] (en = ethylenediamine, tpy = terpyridine, **Ru-1**) and [Ru(Cl-tpy)(dach)Cl][Cl] (dach = 1,2-diaminocyclohexane, **Ru-2**) towards human and murine colon carcinoma cells were tested in vitro and in vivo and compared with oxaliplatin, the most commonly used chemotherapeutic agent against colorectal carcinoma. Ruthenium(II) complexes showed moderate cytotoxicity with IC_50_ values ranging between 19.1 to 167.3 μM against two human, HCT116 and SW480, and one mouse colon carcinoma cell line, CT26. Both ruthenium(II) terpyridine complexes exerted a moderate apoptotic effect in colon carcinoma cells, but induced significant necrotic death. Additionally, both complexes induced cell cycle disturbances, but these effects were specific for the cell line. Further, **Ru-1** significantly reduced the growth of primary heterotopic tumor in mice, similarly to oxaliplatin. Renal damage in **Ru-1** treated mice was lower in comparison with oxaliplatin treated mice, as evaluated by serum levels of urea and creatinine and histological evaluation, but **Ru-1** induced higher liver damage than oxaliplatin, evaluated by the serum levels of alanine aminotransferase. Additionally, the interaction of these ruthenium(II) terpyridine complexes with the tripeptide glutathione (GSH) was investigated by proton nuclear magnetic resonance (^1^H NMR) spectroscopy. All reactions led to the formation of monofunctional thiolate adducts [Ru(Cl-tpy)(en)GS-*S*] (**3**) and [Ru(Cl-tpy)(dach)GS-*S*] (**4**). Our data highlight the significant cytotoxic activity of [Ru(Cl-tpy)(en)Cl][Cl] against human and mouse colon carcinoma cells, as well as in vivo antitumor activity in CT26 tumor-bearing mice similar to standard chemotherapeutic oxaliplatin, accompanied with lower nephrotoxicity in comparison with oxaliplatin.

## 1. Background

Platinum-based drugs such as cisplatin, carboplatin, and oxaliplatin have been among the most effective chemotherapeutic agents in carcinoma treatment for years. However, their high toxicity and the incidence of spontaneous or acquired drug resistance limit their clinical use [1,2,3]. Oxaliplatin is a third-generation diaminocyclohexane (dach) platinum drug that is effective in the treatment of metastatic colorectal carcinoma. Clinically, oxaliplatin induces acute and chronic peripheral neuropathy after prolonged therapy due to a cumulative toxicity, with consequent pain and loss of sensation [4,5]. In the past two decades, ruthenium compounds have attracted considerable interest as potential anticancer agents due to their low toxicity and their efficacy against platinum drug-resistant tumors, reflected in promising results in various stages of preclinical to early clinical studies [6]. Further, the unique properties of ruthenium-based drugs, such as slow ligand exchange rates, range of oxidation states (Ru(II), Ru(III) and Ru(IV)), and favorable water solubility than that of conventional platinum drugs justify the great expectations posed for ruthenium compounds [7]. Three ruthenium compounds, NAMI-A ((ImH)[*trans*-RuCl_4_(dmso-*S*)(Im)], Im (imidazole), dmso (dimethylsulfoxide)), KP1019 (IndH)[*trans*-RuCl_4_(Ind)_2_], Ind (indazole)), and KP1339 (Na[trans-RuCl_4_(Ind)_2_]) have entered human clinical trials [8]. Clinically investigated ruthenium complexes, KP1019 and its sodium salt analogue KP1339, which are active against colon carcinomas, were thought to exhibit tumor selectivity via HSA (Human Serum Albumin) mediated pathways based on increased permeability and the retention effect of tumor tissues. This effect was attributed to the altered structure of tumor blood vessels that allows macromolecules, such as a drug-HSA complex, to pass through gaps in endothelial cells of blood vessels and accumulate in tumor tissue [9].

The synthesis of ruthenium(II) terpyridine complexes [Ru(Cl-tpy)(en)Cl][Cl] (abbreviated **Ru**-**1**) and [Ru(Cl-tpy)(dach)Cl][Cl] (abbreviated **Ru-2**), their stability in aqueous solution, as well as their interaction with different biomolecules were thoroughly investigated [10]. It was demonstrated that these complexes can bind to HSA with moderate-to-strong affinity [11], which may be relevant given the fact that HSA is the leading protein responsible for drug transport and uptake, but is also thought to be crucial for antitumor activity of several ruthenium-based drug candidates [9]. Furthermore, it was reported that ruthenium(II) terpyridine complexes bind to DNA through intercalation by inserting the planar terpyridine ring between the DNA base pairs and through the covalent binding to guanine N7 [12]. Additionally, we reported that there is a relationship between the lipophilicity and citotoxic potency of the ruthenium complexes. We assumed that the most lipophilic Ru complex caused the strongest cytotoxic effect on the tested cells [13]. In the case of complexes **Ru-1** and **Ru-2**, both gave negative log *P*_o/w_ values, showing them to be hydrophilic in nature. Complex **Ru-1** tended to be less hydrophilic (−1.33) compared to complex **Ru-2** (−1.45), which may facilitate its cell uptake efficiency and enhance its anticancer activity [12].

Recent findings show that Ru(II) anticancer drugs have strong affinity for the thiolate sulfur of cysteine and glutathione [14,15]. Drug development involving ruthenium complexes shifted from DNA targeting towards protein targeting drugs. The strength of the interaction with proteins determines whether the drug is inactivated by binding to the plasma protein (detoxification function) or can be released from this adduct under certain conditions (transport function) [16]. Keeping that in mind, in the present work we studied the interactions of Ru(II) terpyridine complexes **Ru-1** and **Ru-2** with the tripeptide glutathione (GSH) by NMR spectroscopy.

Since the potential in vivo antitumor effects of [Ru(Cl-tpy)(en)Cl][Cl] (**Ru**-**1**) and [Ru(Cl-tpy)(dach)Cl][Cl] (**Ru-2**) complexes, (Figure 1), were not reported, we investigated the effects of **Ru-1** and **Ru-2** on colon carcinoma models in vitro and in vivo, as well as evaluated their systemic toxicity in vivo.

## 2. Results

### 2.1. Interactions of Ruthenium(II) Terpyridine Complexes ***Ru-1*** and ***Ru-2*** with Glutathione

The tripeptide glutathione (γ-l-Glu-l-Cys-Gly; GSH) is highly abundant in cells at millimolar concentrations and is well known to be involved in the deactivation of the clinical drug cisplatin and in platinum resistance. In 2002, Sadler et al. reported that ruthenium(II) arene anticancer complexes bind to sulfur-containing amino acids such as l-cysteine and l-methionine forming S-bound adducts [14]. Furthermore, it was demonstrated that GSH is kinetically competitive with guanine (as guanosine 3′,5′-cyclic monophosphate, cGMP) for coordination with ruthenium(II) arena complexes producing a ruthenium thiolate adduct which can be subsequently oxidized by dioxygen to create a unique sulfenate intermediate. These results revealed a facile route for the formation of the thermodynamically stable cGMP adduct via the displacement of *S*-bound glutathione by Guo-N7 [15]. Additionally, Sadler et al. revealed a potentially contrasting role for GSH in the mechanism of the action of the ruthenium(II) arene anticancer complexes that may contribute to the lack of cross-resistance with platinum drugs, a potential clinical advantage [17].

In recent years, we have studied in detail the interactions of Ru(II)-tpy complexes with different amino acids and proteins. In 2019, Rilak Simović et al. reported a systematic review about the chemistry and reactivity of ruthenium(II) complexes and explained the importance of these interactions that help us to reveal the chemical transformations that a metallocomplex undergoes in different biological contexts: cell culture media, circulation, cytoplasm, or the cell membrane before it reaches its biological target (protein(s) or DNA) [13].

In the present work, in order to gain more information on the complexes’ bioavailability we studied the interactions of ruthenium(II) terpyridine complexes **Ru-1** and **Ru-2** with reduced glutathione (GSH) by ^1^H NMR spectroscopy in D_2_O at ambient temperature. The chloride ligand in the cationic compounds **Ru-1** and **Ru-2** proved to be very labile in aqueous solution. Immediately after dissolution in D_2_O, a new set of resonances were observed to grow both in the aromatic (Cl-tpy resonances) and in the upfield (en or dach resonances) regions of the ^1^H NMR spectra. These new resonances, which grew at the expense of those of the parent compound, were attributed to the aqua species [Ru(Cl-tpy)(en)(OH_2_)]^2+^ (1aq) and [Ru(Cl-tpy)(dach)(OH_2_)]^2+^ (2aq), respectively (Scheme 1).

The addition of GSH to an equilibrated solution of **Ru-1** (10 mM) in D_2_O induced very slow changes in the ^1^H NMR spectrum (Figure 2). A new set of resonances, attributable to the thiolate adduct [Ru(Cl-tpy)(en)(GS-*S*)] (**3**), became apparent in the ^1^H NMR spectrum after 6 h. Similarly, the reaction of **Ru-2** with GSH (1:1, 10 mM, D_2_O) yielded one final product that was identified by ^1^H NMR spectroscopy (Figure 3) as the *S*-bonded neutral species [Ru(Cl-tpy)(dach)(GS-*S*)] (**4**).

The NMR time course allowed us to draw the reaction pathways as shown in Scheme 2. GSH binds to 1aq and 2aq through the sulfur atom, giving the corresponding thiolate adducts [Ru(Cl-tpy)(en)(GS-*S*)] (**3**) and [Ru(Cl-tpy)(dach)(GS-*S*)] (**4**).

### 2.2. Ruthenium(II) Terpyridine Complexes Exerts Cytotoxic Capacity against Colon Carcinoma Cells

The MTT assay procedures were used to evaluate the in vitro cytotoxicity of complexes **Ru-1** and **Ru-2** on a mouse carcinoma cell line (CT26) and two human carcinoma cell lines (HCT116 and SW480). Oxaliplatin was included for comparison as a reference substance, and was tested under the same conditions. The above cell lines were treated with concentrations in range 2.3–300 μM of complexes **Ru-1**, **Ru-2,** and oxaliplatin for 24 and 72 h. The IC_50_ values for complexes **Ru-1**, **Ru-2**, and oxaliplatin against these cell lines are summarized in Table 1.

Based on the IC_50_ values, the highest activity of complexes **Ru-1** and **Ru-2** was noticed towards HCT116 tumor cells. The complex **Ru-2** showed lower cytotoxicity than complex **Ru-1** and oxaliplatin. The IC_50_ values of complexes **Ru-1** were slightly lower toward CT26 cells and twice were lower toward SW480 cells than the IC_50_ values of oxaliplatin after 24 h exposure, indicating a higher cytotoxic effect of complex **Ru-1** on CT26 and SW480 than oxaliplatin under identical conditions. Interestingly, comparing the IC_50_ values obtained after 24 and 72 h of treatment it was noticed that **Ru-1** exerts higher cytotoxicity toward CT26 and SW480 cells in comparison with oxaliplatin after 24 h of treatment, while the cytotoxicity of oxaliplatin is stronger 72 h after treatment, indicating that **Ru-1** exerted its cytotoxicity faster than oxaliplatin (Figure 4 and Table 1).

The LDH test showed that ruthenium(II) terpyridine complexes exhibited cytotoxic activity only at higher concentrations (150 and 300 μM). The results revealed that the level of LDH release was higher from CT26 cells treated with ruthenium(II) terpyridine complexes for 24 h compared to the cells treated with oxaliplatin, indicating that ruthenium(II) terpyridine complexes could affect the cell membrane integrity (Figure 5). Additionally, ruthenium(II) terpyridine complexes increased the release of LDH in a dose-dependent manner. The LDH levels increased from 8.37% (HCT116) to 36.43% (CT26) following **Ru-1** treatment and from 2.35% (HCT116) to 33.07% (SW480) following **Ru-2** treatment in comparison with 0% (HCT116) to 2.20% (SW480) after oxaliplatin treatment at a concentration of 300 μM (Figure 5).

### 2.3. Effects of Ruthenium(II) Terpyridine Complexes on Apoptosis in Human and Murine Colon Carcinoma Cells

To investigate the possible apoptotic death of tumor cells treated by complexes **Ru-1** and **Ru-2**, an Annexin V/PI staining assay was performed (Figure 6). All carcinoma cells were cultured in media containing ruthenium complexes or oxaliplatin (IC_50_ concentrations) or in media alone (control).

Apoptosis, or programmed cell death, is a highly regulated process that is limited to individual cells, and does not cause damage to the surrounding cells, hence the apoptosis induction is the most effective method for carcinoma treatment [18]. The obtained data showed that both ruthenium(II) terpyridine complexes as well as oxaliplatin moderately induced apoptotic death of CT26 tumor cells (Figure 6A,B). Further, **Ru-1** and oxaliplatin induced late apoptosis in HCT116 cells, more efficiently than **Ru-2**, while the percentage of early apoptotic HCT116 cells did not differ significantly between treated and untreated cells (Figure 6A,C). However, the highest percentage of early apoptotic SW480 cells was observed after oxaliplatin and **Ru-2** treatments in comparison to untreated cells, while the percentage of late apoptotic SW480 cells did not differ significantly between treated and untreated cells (Figure 6A,D).

### 2.4. Effects of Ruthenium(II) Terpyridine Complexes on Cells Cycle in Human and Murine Colon Carcinoma Cells

Both the apoptosis induction and/or cell cycle arrest may reduce the viability of carcinoma cells [19]. Thus, we next investigated the impact of ruthenium(II) terpyridine complexes on possible cell cycle disturbances in colon carcinoma cells. Consequently, the cell cycle profile of CT26, HCT116, and SW480 cells was determined after exposure to IC_50_ doses of ruthenium(II) terpyridine complexes or oxaliplatin for 24 h.

As shown in Figure 7A, the treatment of CT26 cells with the **Ru-1** complex and oxaliplatin caused an obvious increase in the percentage of cells in G2/M phase, accompanied by a corresponding reduction in the percentage of cells in G0/G1 phase, indicating the induction of G2/M phase arrest by **Ru-1** complex and oxaliplatin. **Ru-1** caused G2/M phase arrest of HCT116 cells (Figure 7B), and **Ru-2** caused G2/M and S phase arrest in treated SW480 cells compared to respective untreated cells (Figure 7C).

### 2.5. Ruthenium(II) Terpyridine Complexes Reduce the Tumor Growth In Vivo

In view of the proven cytotoxic effects of ruthenium(II) terpyridine complexes in vitro, the next goal of the present study was to examine the ability of tested compounds to inhibit murine colon cancer growth and progression in vivo. CT26 cells were heterotopically implanted into the shaved flank of the mice. After the appearance of a palpable tumor, exactly on day six, the treatment of mice began with ruthenium(II) terpyridine complexes (**Ru-1** and **Ru-2**), oxaliplatin, or vehicle (0.9% NaCl) (Scheme 2).

The short-course treatment with ruthenium(II) terpyridine complexes and oxaliplatin (2 mg/kg body weight/every three days, four times in total), started from day six, after the tumor cell implantation, and was associated with the reduction of colon carcinoma growth compared with vehicle treated animals (Figure 8).

Until day 15, both ruthenium(II) terpyridine complexes exhibited similar effects on tumor growth as oxaliplatin. On the last day of the experiment, oxaliplatin and **Ru-1** achieved an approximate effect in inhibiting tumor growth, which was 51.39% and 53.41%, respectively, while **Ru-2** inhibited tumor growth by only 17.56% (Figure 8A). Furthermore, the tumor weights measured after necropsy, were markedly lower in mice treated with **Ru-1** and oxaliplatin in comparison to vehicle-treated mice (*p* = 0.046; *p* = 0.039, Mann–Whitney U test) (Figure 8B), while tumors of mice treated with **Ru-2** were slightly smaller than tumors from the control group, but without statistical significance (Figure 8).

### 2.6. Ruthenium(II) Terpyridine Complexes Are Well Tolerated In Vivo

Finally, in order to examine the safety of ruthenium(II) terpyridine complexes in vivo, the mice’s body weight was measured before tumor cell inoculation and at the end of experiment. The toxicity associated with ruthenium(II) terpyridine complexes treatment was assessed at biochemical and histopathological levels. The serum concentrations of the biochemical markers ALT, AST, urea, and creatinine were obtained to evaluate the liver and renal functions. In addition, the histopathological changes in the target organs, liver, heart, lungs, and kidneys were evaluated.

All mice survived until the end of the study, with no evidence of severe general toxicity. From the first day to the 18th day of the experiment, there were no significant changes in the body weight of mice in all groups (Figure 9A, *p* > 0.05, Student’s *t* test). Also, there were no significant changes in the relative weights of isolated organs (heart, liver, lungs, and kidney) between the groups (Figure 9B, *p* > 0.05, Student’s *t* test).

The group of mice treated with **Ru-2** had slightly higher urea concentration (~3 units) compared to all other groups, but still statistically significant (significant difference from: **Ru-1** group (*p* = 0.008); oxaliplatin group (*p* = 0.010); control group (*p* = 0.017); Figure 9C), while there was no significant difference in creatinine concentration between the groups (Figure 9C). **Ru-1** and oxaliplatin treatment did not affect the serum concentration of urea and creatinine compared to saline-treated mice (Figure 9C).

The serum level of ALT was significantly higher in the group of **Ru-1**-treated tumor-bearing mice in comparison with **Ru-2** and oxaliplatin-treated and control mice (significant difference from: **Ru-2** group (*p* = 0.015); control group (*p* = 0.018); Figure 9C). There was no difference in the values of ALT and AST between the groups that received intraperitoneally **Ru-2** or oxaliplatin compared to the control group (Figure 9C).

No significant histological changes in the heart tissue were observed in **Ru-1** and **Ru-2**-treated mice in comparison with oxaliplatin and saline-treated mice (Figure 10A).

Changes in the kidney tissue found in all treatment groups were hypercellularity of glomeruli with infiltration of lymphocytes and monocytes and parenchymatous degeneration of tubule epithelium with necrosis. In addition to the changes previously mentioned, interstitial bleeding was present at group treated with oxaliplatin; hyaline cylinders and focal hyaline glomerular change at group treated with **Ru-1** and hyaline cylinders and global hyaline alteration and glomerular necrosis at group treated with **Ru-2** (Figure 10B).

A cross section of the lungs showed the presence of lung congestion with rupture of the alveolar septum, perivascular, and peribronchial mononuclear infiltration in all treated groups (Figure 10C).

The liver changes found in all treatment groups were passive hyperemia–congestion; hydrops and balloon degeneration of hepatocytes; focal and confluent necrosis of the middle and peripheral port areas of the lobules (Figure 10D).

There were fields of emphasized geographic-type necrosis in the primary tumors of oxaliplatin-treated mice that were no greater than the necrosis field in untreated mice tumors. The largest part of the primary tumor of **Ru-1**-treated mice was necrotic. Fields of necrosis in the tumors of mice treated with **Ru-2** were larger compared to necrosis fields in the tumors of untreated mice and oxaliplatin-treated mice, but were smaller compared to necrosis fields in the tumor of **Ru-1**-treated mice (Figure 10E).

## 3. Discussion

In the present study, the cytotoxic effects and antitumor activity of two ruthenium(II) terpyridine complexes were compared to oxaliplatin (gold standard therapeutic for the treatment of colorectal carcinoma) [20] towards colon cancer cells in vitro and in vivo, to determine antitumor potential of ruthenium-based drugs in the therapy of colon carcinoma.

**Ru-1** exhibited similar cytotoxic capacity after 24 h exposure compared to oxaliplatin, especially against mouse colon carcinoma cell line (CT26) and human colon carcinoma cell line (SW480) (Table 1, Figure 4). Similar results were obtained in a previously conducted study in which both ruthenium(II) terpyridine complexes, [Ru(Cl-tpy)(en)Cl]Cl (**Ru-1**) and [Ru(Cl-tpy)(dach)Cl]Cl (**Ru-2**), showed moderate to high in vitro cytotoxicity against human cervix carcinoma cell line (HeLa) and human lung carcinoma cells (A549), and moderate cytotoxicity against normal cell line (human fetal lung fibroblast cells (MRC-5), with IC_50_ in the range of 32.80–66.30 μM for **Ru1** and 72.80–110.80 μM for **Ru-2**, respectively [12,21]. These data implicate that **Ru-1** might be considered as a valuable candidate for anticancer therapy.

The release of the LDH enzyme as a biomarker suggests the loss of membrane integrity, indicating on necrosis of cell [22,23]. The results obtained from the LDH assay suggest that ruthenium(II) terpyridine complexes, as opposed to oxaliplatin, influence the integrity of the cell membrane as an alternative way of inducing cell death. The LDH levels increased up to 36.43% for CT26 cells following **Ru-1** treatment, and up to 33.07% for SW480 cells following **Ru-2** treatment at a concentration of 300 μM and only 2.20% for SW480 cells after 24 h treatment with oxaliplatin at the same concentration, which is in line with previous studies [24]. These results are in accordance with previous reports that ruthenium(II) polypyridyl complexes at lower concentrations do not affect the release of LDH enzyme from various tumor cells (HeLa, PC3, LanCap, MCF-7, and MD-MBA 231), but when the cells were treated with higher concentrations of ruthenium(II) polypyridyl complexes, the LDH activity in the culture media increased significantly [25]. However, the percentages of necrotic cells remain generally low after exposure of colorectal tumor cells to piplartine and arene ruthenium(II) complexes [26,27].

Cells were analyzed using flow cytometry using an Annexin-V and PI staining to understand the mode of cell death induced by ruthenium(II) terpyridine complexes. Apoptotic cells that lose asymmetry of their membrane phospholipids leave phosphatidylserine behind the outer leaflet of the plasma membrane. Annexin V, a calcium-dependent phospholipid-binding protein with a high affinity for phosphatidylserine, is used as a sensitive probe for the presence of phosphatidylserine on the cell membrane and hence as a marker of apoptosis. PI is a nonspecific DNA intercalating agent, which is excluded by the plasma membrane of living cells, and thus can be used to distinguish necrotic cells from apoptotic and living cells by supravital staining without prior permeabilization. Since Annexin V-FITC staining precedes the loss of membrane integrity that accompanies the later stage identified by PI, Annexin V-FITC positive and PI negative staining indicates early apoptosis, while viable cells are Annexin V-FITC negative and PI negative. Cells that are in late apoptosis, or already dead cells, are both Annexin V FITC and PI positive [20]. The highest percentage of late apoptotic SW480 cells was observed upon oxaliplatin treatment relative to the untreated but also to the cells treated with ruthenium(II) terpyridine complexes (Figure 6). The previous research indicates that ruthenium(II) complexes have different effects on the induction of apoptosis in SW480 cells, depending on the ligands in their structure, from a very strong impact where 72% cells were in apoptosis, to moderate, with only 31% apoptotic cells [27]. In terms of the results related to the ability of ruthenium(II) complexes to induce apoptosis of HCT116 cells, there are results from the previous research that confirm that ruthenium(II) complexes increase the early and late apoptosis of HCT116 in a time- and concentration-dependent manners, but less than oxaliplatin, which is consistent with our results [26,28,29].

Apoptosis and the arrest of the cell cycle are two processes which are related in different ways. One of the most important targets for anticancer drugs is the regulation of the cell cycle, in particular, the arrest of the cell cycle in G1 and G2 phases plays a crucial role in the development of the cell cycle [30]. It is well known, for instance, that the phase arrest of the G2/M cell cycle is one of the possible mechanisms for apoptosis induction [31]. On the other hand, the arrest of the G0/G1 phase usually stops the cells from proliferating, but offers cells the possibility of repairing the defects caused by anticancer agents [32].

Many ruthenium(II) complexes can induce apoptosis by different antiproliferative mechanism, whether inducing G0/G1, S or G2/M phase arrest [29,33]. **Ru-1** induced the arrest of CT26 and HCT116 cells in the G2/M phase, and **Ru-2** caused G2/M phase arrest in SW480 cells, which is likely related to the molecular changes in the cancer cells (Figure 7). Investigation of the effects of different treatment regimens on the apoptosis of CT26 and HCT116 cells demonstrated that oxaliplatin treatment appears to exhibit moderate effects on cell apoptosis, which is consistent with our research [26,34,35]. There are no in vitro alternative methods or their combinations which could fully replace in vivo methods for acute systemic toxicity [36]. So, in the next segment of our study, we endeavored to discover the effects of ruthenium complexes on mouse colon carcinoma in vivo and eventual systemic toxicity. The body weight and relative organ weight of mice did not indicate to systemic toxicity of ruthenium(II) terpyridine complexes.

**Ru-1** reduced the colon carcinoma growth and progression as evaluated by significantly lower tumor volume and weight (Figure 8). It should be noted that until the 15th day of the experiment, both ruthenium(II) terpyridine complexes exhibited similar effects on the tumor growth as oxaliplatin, after that point, **Ru-1** showed equal effects to oxaliplatin, while **Ru-2** had less suppressed tumor growth in vivo (Figure 8).

Since it has been reported that oxaliplatin would cause liver damage [37], we tested ALT and AST levels in mice serum as indicators for liver function. In our investigation, no substantial changes of blood urea nitrogen, creatinine, aspartate aminotransferase, or alanine aminotransferase were observed and all values were within the normal range in the group of mice treated with oxaliplatin. In similar research, mice bearing CT26 colon tumors were treated with 10 mg/kg oxaliplatin intratumorally, had no elevated biochemical parameters in serum, and no visible treatment toxicity to vital organs according to the hematoxylin-eosin staining of heart, liver, spleen, lung, and kidney sections [38]. However, ruthenium complexes are not completely free of toxicity. As such, ruthenium complexes, such as NAMI-A, can damage the kidneys, leading to dilated tubules and injured glomeruli and elevated levels of serum creatinine in animal assays [39,40]. The main target organ for toxic effects from KP1019 beside kidneys is bone marrow [41].

The histopathological evaluation is considered to be the primary assay to assess the in vivo toxic potential of applied substances. Histopathological analysis of the heart, liver, lung, kidney, and primary tumors was performed, and reversible and irreversible histopathological changes or injuries were found (Figure 10). This indicates that ruthenium complexes, as well as oxaliplatin cause a toxic damage to these organs. The lack of changes in the level of AST and ALT aminotransferases found in the mice is not in correlation with histopathological changes in the liver morphology.

## 4. Materials and Methods

The compounds [Ru(Cl-tpy)(en)Cl][Cl] (**Ru**-**1**) and [Ru(Cl-tpy)(dach)Cl][Cl] (**Ru**-**2**) were synthesized as reported previously [10]. All other chemicals were used as purchased without further purification. NMR spectra were recorded on a Varian Gemini 200 MHz spectrometer. All chemical shifts were referenced to TSP (trimethylsilylpropionic acid) at d = 0.00. All NMR spectra were run at 295 K.

### 4.1. Cell Lines

CT26 (mouse colon carcinoma cell line), HCT116, and SW480 (human colon carcinoma cell lines) were obtained from the American Type Culture Collection (ATCC) [42]. The cells were maintained in Dulbecco’s Modified Eagle Medium (DMEM) supplemented with 10% fetal bovine serum, 200 mM l-glutamine 10,000 units/mL penicillin, and 10 mg/mL streptomycin (all from Sigma-Aldrich, St. Louis, MA, USA). The cells were cultivated at 37 °C in absolute humidity in an atmosphere containing 5% carbon dioxide (CO_2_).

### 4.2. Preparation of Complex Solution

Stock solution of ruthenium(II) terpyridine complexes for in vitro assays were dissolved in sterile saline at a concentration of 600 mM, and diluted by a nutrient cell medium (DMEM, Sigma-Aldrich, Darmstadt, Germany) to various working concentrations. All solutions were prepared at the day of the treatment of the cells.

### 4.3. Cytotoxicity Assays

In order to determine the cytotoxic activity of selected complexes, MTT (3-(4,5-dimethylthiazol-2-yl)-2,5-diphenyltetrazolium bromide), LDH (lactate dehydrogenase), and Annexin V/PI assays were used.

### 4.4. MTT Assay

The cytotoxicity of two ruthenium(II) terpyridine complexes, **Ru-1** and **Ru-2**, and oxaliplatin (as reference substance) on CT26, HCT116, and SW480 was determined by the MTT assay [43]. Due to its clinical use as an anticancer agent, the cytotoxicity of oxaliplatin against the same cell lines was also examined. The cells were harvested from the culture flasks during the exponential growth phase, counted and 5 × 10^3^ cells/well were seeded into 96-well culture plates. About 24 h later, after the cell adherence, each well was treated with 100 μL of tested complexes, which had been serially diluted two-fold in medium to concentrations ranging from 300 to 2.3 μM. All cells were incubated at 37 °C in an atmosphere containing 5% CO_2_ and at absolute humidity for 24 and 72 h. After incubation, the culture medium was removed from each well and MTT solution (5 mg/mL in phosphate buffered saline (PBS)) was added to each well and cells were incubated 4 h under culture conditions. The cell-free supernatants were suctioned off, and dimethyl sulfoxide (DMSO) (150 μL) and glycine buffer (20 μL) were added to dissolve the formazan crystals. The plates were shaken for 10 min. The optical density of each well was determined at 595 nm using a multiplate reader (Zenyth 3100, Anthos Labtec Instruments GmbH, Salzburg, Austria). Experiments were performed in triplicates and repeated in three independent series. The percentage of cell viability was determined by comparison with untreated controls according to formula: % of viable cells = (E − B)/(S − B) × 100, where B is for the background of the medium alone, S is for the total viability/spontaneous death of untreated target cells, and E is for the experimental well. The IC_50_ values were determined by plotting the percentage viability versus concentration on a logarithmic graph and reading off the concentration at which 50% of cells remained viable relative to the control.

### 4.5. Lactate Dehydrogenase (LDH) Assay

The cytotoxicity of used complexes was examined by an In Vitro Toxicology Assay Kit, Lactic Dehydrogenase based (Sigma-Aldrich, St. Lous, MO, USA). Cells were prepared and treated with complexes in the same manner as for the MTT assay. Additional wells were prepared as high control cells were treated with 1/10 volume of LDH Assay Lysis Solution for 45 min. Cells exposed to medium were used as low controls. After treatment, supernatant (50 μL) was transferred to new plate and incubated with an equivalent volume of Lactate Dehydrogenase Assay Mixture (which was prepared by mixing equal volumes of LDH Assay Substrate Solution, LDH Assay Dye Solution, and 1’ LDH Assay Cofactor Preparation. After incubating the plates in the dark for 30 min at room temperature, reaction was terminated by the addition of 10 μL of 1 N HCl to each well and data were acquired by spectrophotometry at 490 nm. The percentage of dead cells was calculated using the formula [44]:% of dead cells = (exp. value − low control)/(high control − low control) × 100.

### 4.6. Annexin V-Propidium Iodide Double Staining Assay

For the detection of apoptosis, the Annexin V binding capacity of treated cells was examined by flow cytometry using an Annexin V-fluorescein isothiocyanate (FITC) Detection Kit (BD Pharmingen, San Jose, CA, USA) according to the manufacturer’s protocol. CT26, HCT116, and SW480 cells were incubated with the appropriate IC_50_ concentrations (calculated previously by the obtained MTT assay results) of ruthenium(II) terpyridine complexes, **Ru-1**, **Ru-2** and oxaliplatin, or with media alone (control) for 24 h at 37 °C in an atmosphere of 5% CO_2_ and at absolute humidity. Following the incubation, all cells were trypsinized, washed in PBS, centrifuged, and resuspended in 100 μL of ice-cold binding buffer (10× binding buffer: 0.1 M Hepes/NaOH (pH 7.4), 1.4 M NaCl, 25 mM CaCl_2_) at a concentration of 1 × 10^6^/mL. Annexin V-FITC and propidium iodide (PI) were added to the 100 μL of cell suspension and incubated for 15 min at room temperature (25 °C) in the dark. After incubation, 400 μL of 1× binding buffer was added to each tube and stained cells were analyzed within 1 h using a flow cytometer FACS Calibur (BD Biosciences, San Jose, CA, USA). Data were analyzed using FlowJo Software version VX [45]. Measurements were presented as density plots of Annexin V-FITC and PI stainings.

### 4.7. Cell Cycle Analysis

To examine the potential effects of ruthenium complexes on cell cycle disturbances of CT26, HCT116, and SW480 cells, all cells were incubated with the appropriate IC_50_ concentrations of ruthenium complexes and oxaliplatin, or with media alone (control) for 24 h at 37 °C in an atmosphere of 5% CO_2_ and at absolute humidity. Cell cycle analysis was performed with a Vybrant^®^ DyeCycle™ Ruby stain (Thermo Fisher Scientific, Waltham, MA, USA) according to the manufacturer’s instructions. CT26, HCT116, and SW480 cells stained with Vybrant DyeCycle Ruby were analyzed by a Fluorescence-activated cell sorting (FACS) Calibur flow Cytometer (BD Biosciences, San Jose, CA, USA). The cell cycle distribution was analyzed using FlowJo software and the results were presented as histograms [46].

### 4.8. Experimental Animals

Male BALB/c mice of 6–8 weeks of age were used in all experiments. Mice were housed in a temperature-controlled environment (22–24 °C) with a 12 h light–dark cycle and were given standard laboratory food and water ad libitum. All animals were housed in a temperature–controlled environment with a 12 h light–dark cycle and were administered standard laboratory chow and water ad libitum. All experiments were approved (01-8461/2) by, and conducted in accord with, the Guidelines of the Animal Ethics Committee of the Faculty of Medical Sciences of the University of Kragujevac (Kragujevac, Serbia).

### 4.9. Animal Model and Drug Treatment

BALB/c mice bearing syngeneic CT26 mouse colon carcinoma were selected as the test system, as this experimental model has been extensively used for this kind of research in the literature [47]. For heterotopic colon carcinoma model, 1 × 10^6^ CT26 cells suspended in 100 μL of DMEM were injected subcutaneously into the left flank of mice. Tumor-bearing mice were examined every 3 days for tumor development and progression and monitoring body weight. Animals were randomized into 4 groups with 6 tumor-bearing mice that were ear-tagged and followed-up individually throughout the study. The intraperitoneal administration of complexes or saline began on sixth day after post-tumor inoculation. Each drug was administered at doses of 2 mg/kg dissolved in 200 μL saline, twice weekly for four times in total (Scheme 2). Treatment groups were as follows: control (saline), **Ru-1**, **Ru-2**, and oxaliplatin. We decided for a dose of 2 mg/kg because the previously conducted pilot experiment showed that ruthenium(II) terpyridine complexes at a dose of 5 mg/kg reduce tumor growth in CT26 bearing mice, but with a pronounced systemic toxicity compared to oxaliplatin.

### 4.10. Estimation of Heterotopic Colon Carcinoma Growth

The size of primary CT26 colon tumors were assessed morphometrically using electronic calipers in two dimensions. The tumor volume was calculated as follows: tumor volume (mm^3^) = (L × W^2^)/2, where L is the longest and W the shortest radius of the tumor in millimeters. Results were expressed as means of tumor volumes ± SD. The percent tumor growth inhibition (TGI) was determined according to the formula, TGI(%) = (Vc − Vt)/(Vc − Vo) × 100, where Vc − Vt are the median of control and treated groups at the end of the study and Vo at the start.

### 4.11. Toxicity Assessment

To study the potential side effects of ruthenium(II) terpyridine complexes, treated mice were monitored for weight loss. All mice were sacrificed in atmosphere saturated with diethyl ether 72 h after the last dose of tested complexes. Blood samples were collected from abdominal aorta of each mouse in tubes without anticoagulant, and a separated serum was processed and analyzed by the chemistry analyzer Roche Cobas Mira Plus. The levels of serum urea, creatinine, and liver enzymes—aspartate aminotransferase (AST) and alanine aminotransferase (ALT)—were determined to assess the renal and hepatic function. After sacrificed, organ weights of the heart, liver, lungs, and kidneys, as well as the tumors, were determined for each animal, and the relative organ weight was calculated as % body weight. The tissue sections of various organs (heart, liver, lungs, and kidneys) and tumor were isolated for histopathological analysis. The tissues fixed in 4% paraformaldehyde were embedded in paraffin, cut into thin sections and mounted on glass slides. The tissue sections were stained with hematoxylin and eosin for microscopic examination.

### 4.12. Statistical Analysis

Statistical analysis of experimental data included the following basic descriptive statistics: the mean value and standard deviation (SD). For testing the normality of the distribution parameters, the Kolmogorov–Smirnov test was used. To test the statistical significance of the results and to confirm the hypothesis, the following statistical tests were used: Student’s *t* test (parametric test), for dependent and independent variables. A database analysis of the results was performed using the software package SPSS 20 (SPSS Inc., Chicago, IL, USA). A *p* value < 0.05 was considered statistically significant.

## 5. Conclusions

The literature data provided information for a lower reactivity of amino acids and proteins towards ruthenium(II) compounds. This may account for the low toxic side effects on such complexes. On the other hand, the relatively weak binding of amino acids and proteins to these complexes may help the transport and delivery of the latter to cancer cells, and allow some amino acids, peptides, and proteins to serve as drug reservoirs for DNA ruthenation, as has been proposed for cisplatin. In order to receive more information about the possible interactions of ruthenium(II) terpyridine complexes with biologically relevant ligands, we have studied the ligand substitution reactions of two Ru(II) terpyridine complexes, [Ru(Cl-tpy)(en)Cl][Cl] (**Ru-1**) and [Ru(Cl-tpy)(dach)Cl][Cl] (**Ru-2**), with reduced glutathione (GSH). According to the NMR results, it is worth noting that the reactions of complexes **Ru-1** and **Ru-2** with GSH lead to the formation of S-bound thiolate complexes, i.e., [Ru(Cl-tpy)(en)(GS-*S*)] (**3**) and [Ru(Cl-tpy)(dach)(GS-*S*)] (**4**), respectively.

This experimental study indicated that ruthenium(II) terpyridine complexes possess significant in vitro cytotoxic activity against human and mouse colon carcinoma, as well as in vivo antitumor activity in heterotopic CT26 tumor models with reduced kidney damage, but with slightly more pronounced liver toxicity compared to oxaliplatin. Therefore, the goals for further research are to investigate other routes of administration or to find the optimal dose of ruthenium(II) terpyridine complexes that will retain the antitumor effect, but have less toxic effects on the organism.

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
