# Peer review of "Antitumor Activity of Ruthenium(II) Terpyridine Complexes towards Colon Cancer Cells In Vitro and In Vivo"

_molecules, 2020, doi:10.3390/molecules25204699_

Round 1

Reviewer 1 Report

The manuscript of Milovanovic and co-workers deals with the cytotoxic evaluation of two ruthenium complexes through a series of different biological assays. The manuscript resulted well-written (although some typos and changes in the font are present through the paper), clear and definitely interesting, taking into consideration the valuable presence of the in vivo studies. Although a fully consistent biological characterization of the complexes, the manuscript lacks completely a chemical point of view, that it’s mandatory in my opinion for publishing in such a journal as Molecules, especially for a paper submitted to an Inorganic Chemistry section, thus making the paper appear to me quite out of scope respect to the aims of the journal. Indeed, I suggest the authors to introduce a figure reporting the structures of the complexes, even if they have been already published elsewhere, and to add some chemical characterization of the complexes, such as their interaction with glutathione via NMR or their lipophilicity via RP-HPLC in order to give some conclusions about the complexes’ bioavailability (see for example: Eur. J. Med. Chem. 2017, 131,196-206; Angew. Chem. Intern. Ed. 2012, 51, 6742-6747). Another aspect the authors could explore is the photophysical properties of the complexes: did the authors check if the complexes are emitters? Please take into  consideration also the effect that different counter anions could have on this aspect of their chemical behaviour (see for example: Inorganica Chimica Acta 2017, 461, 267-274; J. Med. Chem. 2014, 57, 4906−4915). For these reasons, I recommend the publication in Molecules only after this major revision.

Author Response

I would like to thanks for valuable suggestions. The manuscript has been revised taking into consideration all comments. The alterations made are detailed below and are evidenced in yellow in the manuscript.

I hope that you will now find the content of the present manuscript suitable for publication in Molecules.

The manuscript of Jovanovic and co-workers deals with the cytotoxic evaluation of two ruthenium complexes through a series of different biological assays. The manuscript resulted well-written (although some typos and changes in the font are present through the paper), clear and definitely interesting, taking into consideration the valuable presence of the in vivo studies. Although a fully consistent biological characterization of the complexes, the manuscript lacks completely a chemical point of view, that it’s mandatory in my opinion for publishing in such a journal as Molecules, especially for a paper submitted to an Inorganic Chemistry section, thus making the paper appear to me quite out of scope respect to the aims of the journal. Indeed, I suggest the authors to introduce a figure reporting the structures of the complexes, even if they have been already published elsewhere, and to add some chemical characterization of the complexes, such as their interaction with glutathione via NMR or their lipophilicity via RP-HPLC in order to give some conclusions about the complexes’ bioavailability (see for example: Eur. J. Med. Chem. 2017, 131,196-206; Angew. Chem. Intern. Ed. 2012, 51, 6742-6747).

We have added schematic structures of the Ru(II) terpyridine complexes Ru-1 and Ru-2 and of the tripeptide glutathione (γ-L-Glu-L-Cys-Gly; GSH). It is text line 118, Figure 1 in the revised manuscript.

We have also studied the interaction of Ru-1 and Ru-2 with glutathione by NMR spectroscopy as the reviewer suggested. Text line from 292 to 347, Figures 2 and 3, and Scheme 1 are added in the revised manuscript. The obtained results showed that complexes Ru-1 and Ru-2 bind to GSH through the sulfur atom giving the corresponding thiolato adducts.

As the reviewer suggested, we have added the paragraph (text line from 94 to 100 in the revised manuscript) about the lipophilicity of the studied complexes. In our previous paper Rilak et al., Dalton Transactions, 2016, 45, 4633-4646, we have explained the lipophilicity of complexes Ru-1 and Ru-2. It was determined by measuring the concentration ratio of the corresponding complex in the aqueous phase in the equilibrium state. After mixing with octanol and water, Ru-1 and Ru-2 were distributed mostly in the aqueous phase.

Another aspect the authors could explore is the photophysical properties of the complexes: did the authors check if the complexes are emitters? Please take into  consideration also the effect that different counter anions could have on this aspect of their chemical behaviour (see for example: Inorganica Chimica Acta 2017, 461, 267-274; J. Med. Chem. 2014, 57, 4906−4915). For these reasons, I recommend the publication in Molecules only after this major revision.

In our previous papers, we have thoroughly studied the interaction of ruthenium(II) terpyridine complexes with calf-thymus DNA (CT DNA) employing fluorescence quenching measurements. Additionally, we have conducted a detailed study of their interactions with two major metal-transporting serum proteins, human serum albumin and transferrin by fluorescence quenching. It was demonstrated that they bind strongly to calf thymus (CT) DNA both covalently and non-covalently, intercalating between base pairs (Rilak et al., Dalton Trans., 2016). Furthermore, it was suggested that Ru complexes also bind to human serum albumin from moderate-to-strong (Masnikosa et al., Arab. J Chem., 2018, 11, 291-304). In the revised manuscript we have added the appropriate references.

Reviewer 2 Report

The Scientists present the work titled “Antitumor activity of ruthenium(II) terpyridine complexes towards colon cancer cells in vitro and in vivo” that is adequate to the content of the article. The manuscript fits within the scope of the Molecules journal. The Authors continued the research on the Ru compounds prepared by Rilak et.al. in 2014. The research were well planned and consequently done. They included cytotoxicity measurement of two Ru complexes and referenced oxaliplatin on two human and one murine colon carcinoma cells using MTT and LDH methods, the impact of compounds on apoptosis in above carcinoma cells and on cell cycle distributions. Then the ability to inhibit murine colon cancer growth and progression in vivo was examined. Finally the toxicity of the tested compounds was assessed at biochemical and histopathological levels.

All used methods were clearly explained. The results were presented clearly and correctly. The manuscript included broad and adequate Discussion section. The conclusions or summary are accurate and supported by the content. Congratulations to the Authors for the valuable work.

I have some minor recommendations for the authors:

Please develop the abbreviations en and dach in the Abstract.

It would be nice to have the view into the Scheme of  Ruthenium complexes Ru-1 and Ru-2 structures, eg. in Introduction section. Could you include as the scheme?

Line 211 please write the word “colon” twice time

If possible, please explain why the two human colon carcinoma cells were used. What is the difference between HCT116 and SW480 cells?

Could you explain the decrease of cytotoxic activity of Ru-1 after 72h on CT26 cells?

Is the drastic increase (100 times) of cytotoxicity of oxaliplatin in time (after 72h) typical for this compound? (against SW480)

Fig. 2, 3 and 4 titles without bold

To sum up, I am recommending to accept this valuable research.

Author Response

Dear Sir,

I would like to thank for positive review. We did corrections according the suggestions. The alterations made are evidenced in yellow in the manuscript. We also tried to give the answers to the question.

Please develop the abbreviations en and dach in the Abstract.

We added the these abbreviations in the Abstract.

It would be nice to have the view into the Scheme of  Ruthenium complexes Ru-1 and Ru-2 structures, eg. in Introduction section. Could you include as the scheme?

We added the scheme of Ru-1 and Ru- structures, it Scheme 2 in revised manuscript.

Line 211 please write the word “colon” twice time

It is corrected.

If possible, please explain why the two human colon carcinoma cells were used. What is the difference between HCT116 and SW480 cells?

The genetic background of tumors plays a key role in tumor response to chemotherapeutic agents (Nature. 2013 Dec 12;504(7479):296-300). Response to inhibitors of autophagy depends on type of cell line, but every cell line has the other genetic background (Sci Rep. 2019 Aug 5;9(1):11316). It is possible that two different tumor cell lines have the different response to the same agent and we decided to use two human cell lines and we randomly choosed SW480 and HCT116. We did not have any special reson for choosing right these two cell lines.  

Could you explain the decrease of cytotoxic activity of Ru-1 after 72h on CT26 cells?

We believe that decrease of cytotoxic activity of Ru-1 72 after exposure on CT26 is probably due to the mechanism of induction of CT26 cell death. Ru-1 induces significant release of LDH from CT26 cells while this was not observed in cultures of SW480 and HCT116 cells treated with Ru-1. Release of LDH is marker of necrotic detah (Rayamajhi M et al. Methods Mol Biol. 2013;1040:85-90.). MTT assay measures the mitochondrial metabolic rate and indirectly reflect the viable cell numbers (Cancer Lett. 245, 232–41 (2007). It is possible that Ru-1 quickly exerts its effect on CT26 cells and induce necrosis of certain number of cells, the rest of the cells (viable cells) continue to growth which can be detected as decreased cytotoxic activity after 72 hours mesaured by MTT assay. Significant necrosis in tumor tissue of CT26 colon carcinoma bearing mice treated with Ru-1 (figure 10) support this hypothesis. The other possibility, we believe less likely, is that  Ru-1 complex induces some changes in CT26 mitochondria which become more active and thus influence on the detection of cell viability by MTT assay. Similar observation was noted for radiotherapy (Sci Rep 2018;8(1):1531).

Is the drastic increase (100 times) of cytotoxicity of oxaliplatin in time (after 72h) typical for this compound? (against SW480).

Similar effect of oxaliplatin on SW480 cells has been published previously (Mol Cancer. 2008;7:14. doi: 10.1186/1476-4598-7-14). There almost was no cells in the SW480 cultures after 72 hours of oxaliplatin treatment.

Fig. 2, 3 and 4 titles without bold

we corrected it

Reviewer 3 Report

In this manuscript, authors investigated in vivo and in vitro antitumor activities of water soluble ruthenium complex, [Ru(Cltpy)(L)Cl]. The substitution reaction DNA binding of the complex was previously studied by Rilak et al, and this work successfully expanded the chemistry to colon cancer cells. The experimental results clarified that the complex possesses moderate antitumor activities which are comparable to those of well-known platinum complex. The manuscript is well written; the experimental details are well listed and the results are well explained. Therefore, I recommend the acceptance of the manuscript.

Comments.

1) Authors need to report IC50 and LDH values according to those error bars. For instance, in Table 1, the IC 50 value of CT26 with Ru-1 should be reported as 43 ± 4. Although the error bar seems to be more than 3% for each concentration in Figure 2, the LDH levels were also reported as 4 digits

2) Check the typos in the manuscript. For example,

(a) page 14, line 394, “is be used”

(b) page 13 line 365 “with of”

3) Add abbreviations of FITC and tpy in page 16.

Author Response

Dear Sir,

I would like to thanks for positive review. The corrections are done according the suggestions. The alterations made are evidenced in yellow in the manuscript.

1. Authors need to report IC50 and LDH values according to those error bars. For instance, in Table 1, the IC 50 value of CT26 with Ru-1 should be reported as 43 ± 4. Although the error bar seems to be more than 3% for each concentration in Figure 2, the LDH levels were also reported as 4 digits.

We corrected the IC 50 value according the suggestion.

2. Check the typos in the manuscript. For example,

(a) page 14, line 394, “is be used”

(b) page 13 line 365 “with of”

We review manuscript thoroughly and did our best to correct the typos.

3. Add abbreviations of FITC and tpy in page 16.

We added abbreviations (for tpy in abstract and for FITC on page 8 in revised version of the manuscript

Round 2

Reviewer 1 Report

The authors introduced all the changes requested. I've really appreciated the interaction study between the ruthenium complexes with GSH by NMR.

In this present form, in my opinion, the manuscript results suitable for publication in Molecules.